# Systematic Generalization and Emergent Structures in Transformers Trained on Structured Tasks

## Abstract

Transformer networks have seen great success in natural language processing and machine vision, where task objectives such as next word prediction and image classification benefit from nuanced context sensitivity across high-dimensional inputs. However, there is an ongoing debate about how and when transformers can acquire highly structured behavior and achieve systematic generalization. Here, we explore how well a causal transformer can perform a set of algorithmic tasks, including copying, sorting, and hierarchical compositions of these operations. We demonstrate strong generalization to sequences longer than those used in training by replacing the standard positional encoding typically used in transformers with labels arbitrarily paired with items in the sequence. We search for the layer and head configuration sufficient to solve these tasks, then probe for signs of systematic processing in latent representations and attention patterns. We show that two-layer transformers learn generalizable solutions to multi-level problems, develop signs of systematic task decomposition, and exploit shared computation across related tasks. These results provide key insights into how stacks of attention layers support structured computation both within task and across tasks.

## 1 Introduction

Since their introduction (Vaswani et al., 2017), transformer-based models have become the new norm of natural language modeling (Brown et al., 2020; Devlin et al., 2018) and are being leveraged for machine vision tasks as well as in reinforcement learning contexts (Chen et al., 2021; Dosovitskiy et al., 2020; Janner et al., 2021; Ramesh et al., 2021). Transformers trained on large amounts of data under simple self-supervised, sequence modeling objectives are capable of subsequent generalization to a wide variety of tasks, making them an appealing option for building multi-modal, multi-task, generalist agents (Bommasani et al., 2021; Reed et al., 2022).

Central to this success is the ability to represent each part of the input in the context of other parts through the self-attention mechanism. This may be especially important for task objectives such as next word prediction and image classification at scale with naturalistic data, which benefit from nuanced context sensitivity across high-dimensional inputs. Interestingly, transformer-based language models seem to also acquire structured knowledge without being explicitly trained to do so and display few-shot learning capabilities (Brown et al., 2020; Linzen & Baroni, 2021; Manning et al., 2020). These insights have led to ongoing work exploring these models' potential to develop more broad reasoning capabilities (Binz & Schulz, 2022; Dasgupta et al., 2022).

Despite success in learning large-scale, naturalistic data and signs of acquisition of structured knowledge or generalizable behavior, how transformer models support systematic generalization remains to be better understood. Recent work demonstrated that large language models struggle at longer problems and fail to robustly reason beyond the training data (Anil et al., 2022; Razeghi et al., 2022). Different architectural variations have been proposed to improve length generalization in transformers, highlighting the role of variants of position-based encodings (Csordás et al., 2021a;b; Ontanón et al., 2021; Press et al., 2021). Indeed, whether neural networks will ever be capable of systematic generalization without building in explicit symbolic components remains an open question (Fodor & Pylyshyn, 1988; Smolensky et al., 2022).

Here, we approach this question by training a causal transformer model to perform a set of algorithmic operations, including copy, reverse, and hierarchical group or sort tasks. We explicitly sought the minimal transformer that would reliably solve these simple tasks and thoroughly analyze such minimal solution through attention ablation and representation analysis. Exploring how a transformer with no predefined task-aligned structure could adapt to structures in these algorithmic tasks provides a starting point for understanding how self-attention can tune to structures in more complex problems, e.g., those with the kinds of exceptions and partial regularities of natural datasets, where the exploitation of task structures may occur in a more approximate, graded manner. Our main contributions are:

1. We present a set of two-layer causal transformers that are capable of learning multiple algorithmic operations, and show that putting more attention heads at deeper layers has advantages for learning multi-level tasks.

2. We show that the attention layers in these models reveal signs of systematic decomposition within tasks and exploitation of shared structures across tasks.

3. We highlight a simple label-based order encoding method in place of the positional encoding methods typically used in transformers, and show that it helps to achieve strong length generalization performance.

## 2 METHOD

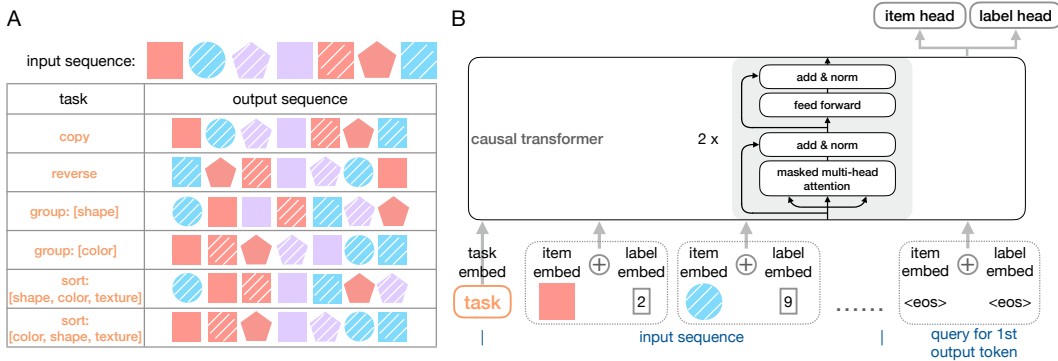

Figure 1: Task and model design.

**Dataset**. We created an item pool covering all combinations of 5 shapes, 5 colors, and 5 textures, and generated a sequence dataset by sampling 100k sequences of 5–50 items randomly selected from the item pool. The tasks we used to train the models are shown in Fig 1A. Each task corresponds to one of the following rules, which relies on item feature and/or item order information to rearrange an input sequence (grouping or sorting items by a particular feature is with respect to a pre-defined feature sort order, e.g., circles < squares < pentagons, or red < purple < blue):

COPY (C): copy the input sequence.
REVERSE (R): reverse the input sequence.
GROUP[SHAPE] (G[S]): group the items by shape, preserve the input order within each shape group.
GROUP[COLOR] (G[C]): group the items by color, preserve the input order within each color group.
SORT[SHAPE,COLOR,TEXTURE] (S[S]): sort the items first by shape, then by color, then by texture.
SORT[COLOR,SHAPE,TEXTURE] (S[C]): sort the items first by color, then by shape, then by texture.

We instantiated the token vocabularies as onehot or multihot vectors. The task tokens were onehot vectors with the corresponding task category set to one, with one additional task dimension corresponding to the end-of-sequence (EOS) token. The item tokens were multihot vectors whose units indicated its value in each feature dimension (equivalent to concatenated onehot feature vectors). As such, the model receives disentangled feature information in the input, though in principle it can learn to disentangle feature information given onehot encodings for each unique item.

**Label-based order encoding**. Using position-based order encodings (whether absolute or relative), models trained with sequences up to length $L$ encounter an out-of-distribution problem when tested

on longer sequences, as position encodings beyond $L$ are unfamiliar to the model. We thus introduce label encoding, which allows longer sequences of items to be encoded with familiar labels. Instead of using item positions, we paired items in each sequence with ascending random integer labels to communicate order information (Fig 1B). The sampled labels were represented as onehot vectors which the models then embed with learnable label weights. In all reported results, we pre-generated item labels sampled from a range up to the maximum generalization length (50) for all sequences in the dataset, and these labels were shared across training steps and model seeds. In practice, the labels for each sequence can be sampled online and from a larger range to accommodate generalization to even longer sequences. We contrast our random label encoding scheme with sinusoidal and learnable encodings based on position indexes.

**Model**. The main model architecture is shown in Fig 1B. Each input sequence consisted of a task token and the paired item and label tokens, with the EOS token serving as the first query for tokens in the output sequence. The input tokens were first embedded to the model's latent representational space through a set of embedding layers depending on the token type (task, item, or label). The item and label embeddings were then added to form a composite item embedding. These embedded tokens were fed into a causal transformer, which contained one or two layers of alternating future-masked attention sublayers and MLP sublayers. Residual connections and layer normalization were applied after each sublayer as in Vaswani et al. (2017). We tested architectural variations in the number of attention heads in different layers of the model while controlling for the total number of learnable parameters (see detailed hyperparameters in Appendix B). The state of the query token at the output of the causal transformer was passed through two linear heads to predict the next output token (the task token, or an item and its associated label).

**Training and evaluation**. The models were trained using full teacher forcing (where we always feed the model the correct input tokens) on all sequences of lengths 5 to 25 in the dataset ($\sim$46k) and evaluated for length generalization on sequences of lengths 26 to 50 ($\sim$54k). We trained models in both single-task and multi-task settings. In both cases, the output sequence consisted of the correctly ordered items and their labels given the task being trained, followed by an EOS token. In single-task learning, we did not include the task token in training or evaluation. In multi-task learning, the task token was used and the models were trained to first output the task token before predicting the output sequence. The training sequences used in multi-task learning remained the same ones between lengths 5–25, but each sequence corresponded to a different output sequence under different tasks.

The models were trained using softmax cross-entropy loss on the prediction of feature classes, labels, and task/EOS categories for tokens in the output sequence. Item predictions were treated as average feature prediction accuracy, i.e., if the model predicted 2/3 features correct, its token-level item accuracy is 2/3. Training stopped at 32k gradient updates for single-task models and 38k gradient updates for multi-task models. Below, we report both token-level and sequence-level accuracy, under both teacher forcing and top1 rollout (i.e., greedy decoding). Results were aggregated over four random seeds for each task type $\times$ architecture pair. Unless otherwise specified, results were taken from the checkpoint with the highest generalization accuracy within each seed. Error shades and error bars indicate standard error of the mean across models.

## 3 RESULTS

### 3.1 SINGLE-TASK LEARNING

**Two-layer models with label encoding learn the SORT task and generalize to longer sequences**. We first trained the model with the SORT[SHAPE,COLOR,TEXTURE] task. Using our label encoding method, models with two single-headed layers (indicated as [1,1]) were able to achieve near-ceiling accuracy on training sequences and generalize to longer sequences (Fig 2; also see quantitative results in Appendix C). The predictions of the EOS token were also highly accurate in these models (see Fig S1A in Appendix A.1). Item prediction was more accurate than label prediction in this task, reflecting that the models represented item feature information more accurately in order to sort the input tokens. The two-layer models showed some degradation in sequence-level accuracy as a function of sequence length, but the failures on longer sequences were not catastrophic, as these models scored very well on longer sequences when up to 5% prediction errors were allowed (Fig 2D; also see Fig S1B, and Fig S2 for accuracy under rollout in Appendix A.1). In contrast,

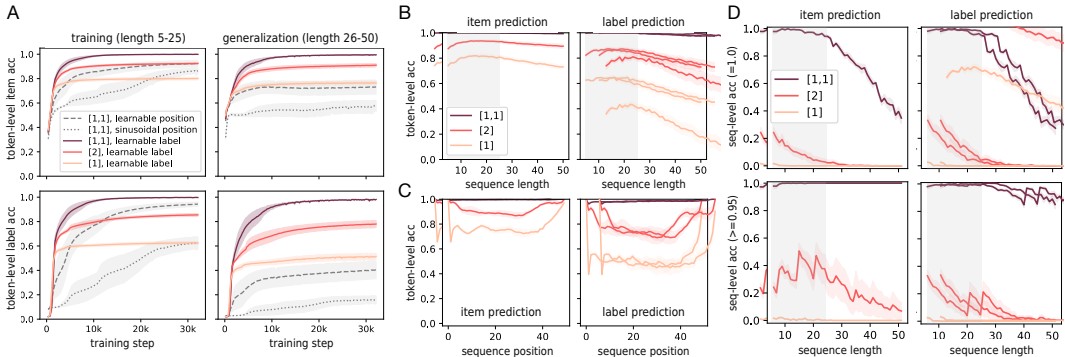

Figure 2: Token- and sequence-level accuracy for single-task models. **A**. Token-level accuracy on training and generalization sequences over learning. **B**. Token-level accuracy over sequence length. **C**. Token-level accuracy over sequence positions. **D**. Proportion of sequences that the model predicted 100% tokens correct (upper) or predicted greater than 95% tokens correct (lower). In B, C, and D, results were taken from 5k training sequences (in B and D) and 5k generalization sequences (B, C, and D). Legends indicate the number of attention heads in each layer and the order encoding used (in A). Gray shades indicate the sequences used in training.

two-layer models trained with sinusoidal or learnable position embeddings performed worse across both training and generalization sequences (Fig 2A).

The two-layer models were also much better than single-layer models with either one or two attention heads. While these single-layer models were able to exploit some correlations between items and output positions (e.g., item [0,0,0] always came first, and item [4,4,4] always came last), they failed to sort items in the middle positions (Fig 2C). In contrast, a single-layer, single-headed model was sufficient to learn the COPY or the REVERSE task (see Fig S3A in Appendix A.1), suggesting that multiple layers strongly benefit successful learning of multi-level problems.

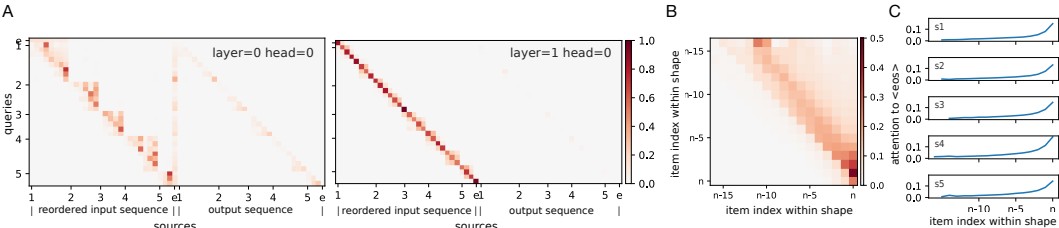

Figure 3: Attention patterns in two-layer models. **A**. Attention maps for an example generalization sequence. Items in the input sequence were reordered to match their output order for visualization purposes. Numbers 1-5 mark the beginning of each shape group. Label e indicates the EOS token. **B**. First-layer attention from query items to input items within shape groups. **C**. Attention to EOS as a function of item index within each shape group (indicated by labels s1-s5). Results in B and C were aggregated across 1k generalization sequences and across seeds.

**Distinct two-stage processing across attention layers**. The attention weights in the two-layer models revealed signs of task decomposition (Fig 3A). The attention head in the first layer tended to distribute attention to the unsorted items that share the same shape as the current query item. The attention head in the second layer then almost exclusively attended to the next output token in preparation for feature and label readout. This pattern appeared robustly across sequences and across different seeds (Fig 3B). Interestingly, there was an increase in the attention weights to the EOS token as the model received query items towards the end of each shape group. This attention to EOS increased to similar degrees in early or late shape groups (Fig 3C), again suggesting that the model learned to systematically process items within each shape group, even though generating the EOS token was only relevant after sorting all items. We also found similar attention patterns in two-layer, single-headed models learning the GROUP[SHAPE] task (see Fig S3B in Appendix A.1).

The single-layer models displayed some attention to subsequent items in the output sequence but lacked consistent structures across different shape groups (see Fig S4 in Appendix A.1). This could be due to the burden for the attention head(s) within a single layer to implement a mixture of item contextualization and readout of the correct item or label, and reflects an advantage of the two-layer models in injecting an inductive bias for a multi-stage solution.

**Acquisition of within-feature order information**. To solve the sort task accurately, the models needed to learn the invariant sort order within each feature. We tested if the learned order information can be parsed out from the input embeddings. We found that the embedding weights associated with shape and color feature units reflected within-feature order similarities, with weights associated with different features appearing mostly orthogonal (Fig 4A). The weaker structure and smaller magnitude associated with texture weights may be due to lower demand in sorting multiple texture values, as texture was the third-level sort feature and thus had fewer consecutive values in a single output sequence under limited sequence length.

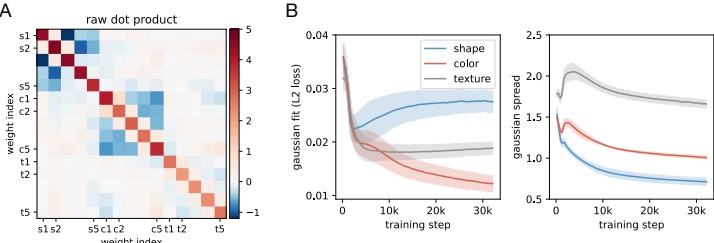

Figure 4: Input item embeddings capture information about feature sort order. **A**. Dot product similarity measure of item embedding weights. **B**. Fitting Gaussian representations to track learned feature sort order in the input embeddings. Left, fit objective (mean L2 loss) over pairwise similarities of feature embeddings. Right, best-fitting Gaussian spread parameter for each feature.

We quantitatively tracked how information about feature sort order was acquired in the input embeddings over learning, using an L2 loss over the difference in the pairwise similarities from synthetic Gaussian feature representations and that from the models' feature embeddings. This analysis suggested that the two-layer models initially began to acquire feature sort order information concurrently in all three features (Fig 4B, left panel). In later stages during learning, the sort order information in color embeddings more quickly and better corresponded to similarities between Gaussian representations, with shape embeddings deviating from Gaussian-like, monotonic order similarities.

## 3.2 MULTI-TASK LEARNING

**Multi-task learning and length generalization in two-layer, multi-headed models**. To explore the ability for the causal transformer to simultaneously learn multiple algorithmic tasks, we trained models to predict different output sequences on the same input sequence conditioned on a task token. The two-layer, single-headed model used in single-task learning was unable to learn all six tasks, while two-layer, multi-headed models achieved good training and generalization performance (Fig 5A, also see quantitative performance in Appendix C). Increasing the number of attention heads in the model did not lead to drastically different learning curves under teacher forcing, but more attention heads supported much better performance under top1 rollout (Fig 5B).

We tested whether multi-headed attention served for better learning when it occurred in the first layer (attention-frontload) or the second layer (attention-backload). Attention-backload models generally achieved higher accuracy across all tasks compared to their attention-frontload counterparts (Fig 5C and 5D). Performance of the attention-backload models was even comparable with their attention-balanced counterparts, which signals that a bottlenecked architecture may be particularly suited for multi-task learning in our setting, considering that some tasks in the task suite share the first-level grouping feature.

Consistent with the single-task model, six-task models demonstrated strong generalization to longer sequences (Fig 6A). Even though sequence-level accuracy degraded as sequence length increased, the models only made less than 5% prediction errors for long extrapolation sequences (Fig 6C). Token-level accuracy among generalization sequences was also relatively stable until the last few

output positions (Fig 6B). We also observed that accurately predicting item features in long sequences was easier in the SORT tasks but harder in the COPY, REVERSE, and GROUP tasks (Fig 6A and 6C). This echos results from the single-task models and again suggests that the models more accurately represented information directly used for sorting the items in each task. See additional results on EOS prediction and accuracy under rollout in Appendix A.2.

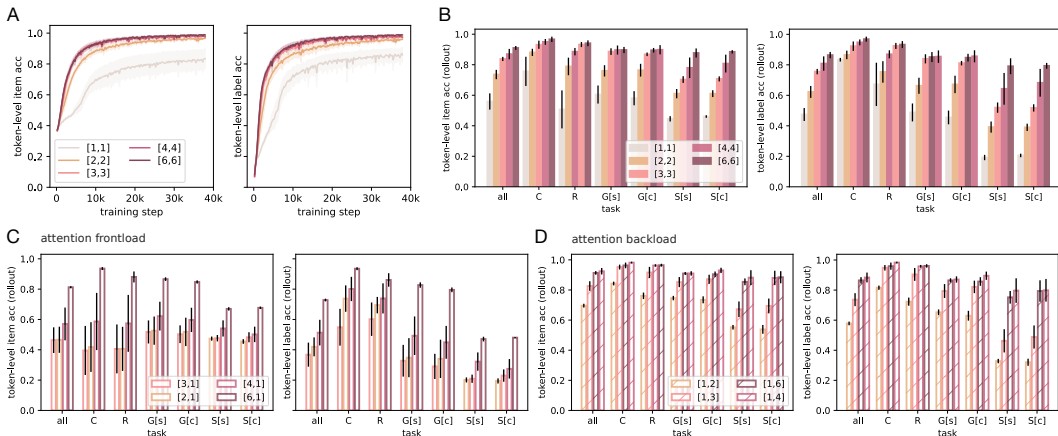

Figure 5: Token-level item and label accuracy in multi-task models. Legends indicate the number of attention heads in the two layers. **A**. Accuracy across all tasks over learning. **B**, **C**, and **D**. Top1 rollout accuracy for models with attention-balanced, attention-frontload, and attention-backload architectures. In A, results were taken from 75k generalization sequences (12.5k per task). In B, C, and D, results were taken from 1.2k generalization sequences (200 per task).

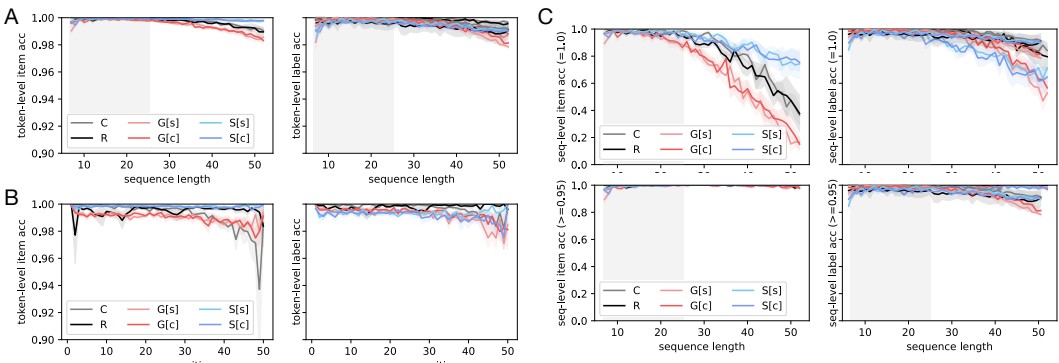

Figure 6: Length generalization across tasks. **A** and **B**. Token-level item and label accuracy over sequence length and sequence position. **C**. Proportion of sequences that the model predicted 100% tokens correct (upper) or predicted greater than 95% tokens correct (lower). Results were taken from 1k training sequences (in A and C) and 1k generalization sequences (in A, B, and C) for each task. Performances were aggregated across the five top-performing runs. Gray shades indicate the sequences used in training.

**Learned task embeddings recover task similarity.** We examined the input task embeddings to understand the basis of task-shared and task-specific computation in the model. Fig 7A shows that similarities among the learned task embeddings reflected shared computation across tasks. For example, the COPY task was recognized to be similar to the REVERSE task and the two GROUP tasks, potentially reflecting the shared stronger attention to label information in order to accurately sort the input items. Representations for the two SORT tasks were also highly similar, as they both rely on item features to sort the input items. The models also recovered similarities between the pairs of GROUP and SORT tasks that share the same first-level grouping feature.

**Task-shared and task-dependent computation across attention layers.** Structures in the task embedding similarities and strong performance from models with fewer attention heads than there

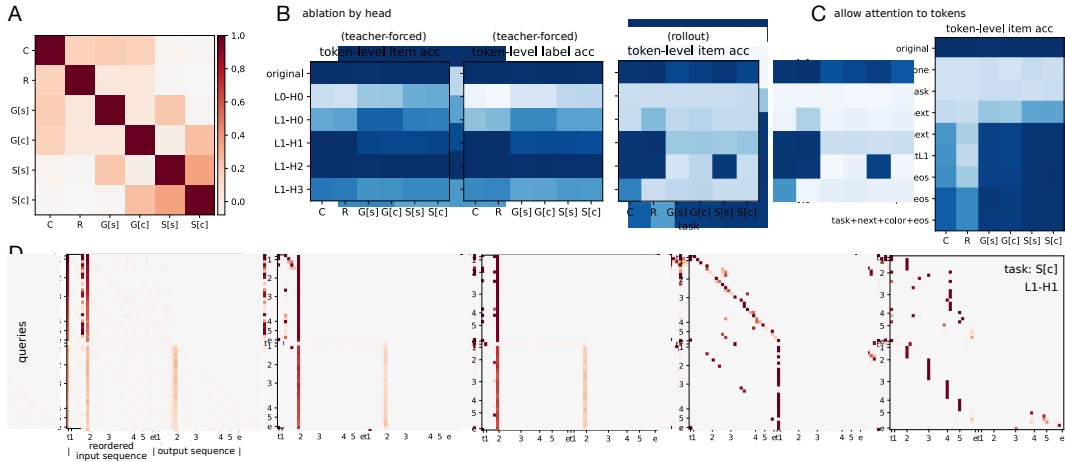

Figure 7: **A**. Pairwise similarities of task embeddings. **B**. Token-level accuracy from ablating single attention heads. **C**. Token-level accuracy from preserving attention to certain tokens across attention heads. **D**. Attention maps from selected attention heads in an example generalization sequence (see Fig S7 for all attention maps). Items corresponding to the input sequence were reordered to match their output order for visualization purposes. Numbers 1-5 mark the beginning of each first-level feature group (shape or color). Label t indicates the task token. Label e indicates the EOS token. In B, C, and D, results were taken from the top-performing model (architecture: [1,4]).

are tasks already hinted that the model could be exploiting shared processing across tasks. Because task-conditioned computation can only occur across the attention layers, we next sought to understand the role of multi-headed attention in implementing such shared computation. We performed two ablation experiments: ablating a single attention head entirely and preserving attention in all other heads, or preserving attention to certain tokens across all attention heads. Attention weights were ablated by masking the attention weights as zero after softmax.

Fig 7B and C show the ablation results for the top-performing model with one attention head in the first layer and four attention heads in the second layer. The attention heads did not exhibit strong selectivity for single tasks as they usually contributed to multiple tasks, and they also showed equal contribution to item and label prediction (Fig 7B, also see results from other models in Fig S8 in Appendix A.2). One attention head appeared redundant as ablating it resulted in little impact on the performance of any task under teacher forcing, but it significantly improved accuracy under top1 rollout.

Ablating attention to certain items in the sequence further indicated that the models were heavily relying on the learnable task embedding to contextualize items under different tasks (Fig 7C, also see Fig S8). When the models were only allowed to attend to the task token in the first layer and the next output token in the second layer, performance was largely preserved in the GROUP tasks and the SORT tasks (ablation type "taskL0+nextL1"). This is different from the two-layer single-task models, whose first layer relied on attention to other items in the first-level feature group to cross-contextualize items. However, there were still some signs of task decomposition in the multi-task model. For example, one attention head in the second layer of the top-performing model consistently attended to the first item in the next feature group across all multi-level tasks (Fig 7D; see detailed attention maps in Fig S7 in Appendix A.2), which proved useful for accurate predictions across multi-level tasks under both teacher forcing and rollout (Fig 7B).

**Task-conditioned item encoding.** To better understand the effect of task-conditioned computation on encoded item representations, we analyzed how the items were represented under different tasks after each layer. We computed average item representations of all 25 shape × color feature pairs in each task, aggregating across texture and label values in 200 generalization sequences. Projecting these task-conditioned item representations to lower dimensions revealed different embedding structures across the two layers (Fig 8). For the two GROUP tasks, representations in the first layer encoded all items by their relevant first-level grouping feature (shape or color). The item representations

did not consistently encode other feature information across both layers, presumably because label information became more relevant for sorting items within each feature group.

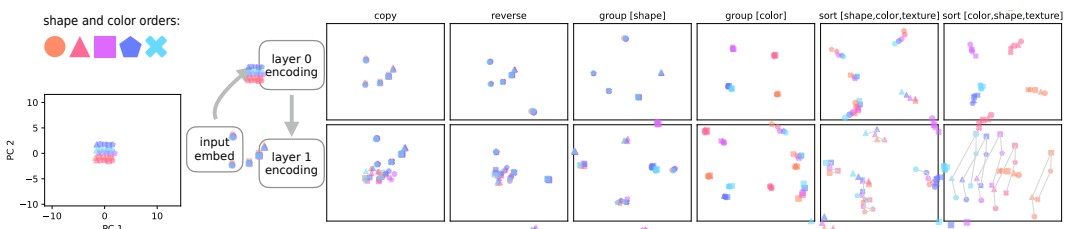

Figure 8: Task-conditioned encoding of the same sequence of items from the top-performing model. PCA was performed for average item representations across 200 generalization sequences within each layer × task pair. The legend indicates the sort order of the shape and color features.

Interestingly, for the two SORT tasks, the encoded representations after the first layer also showed a high degree of clustering in the corresponding first-level grouping feature, with the secondary sort feature represented in a ring structure. The subsequent layer then expanded the item representations along the secondary feature in a way that was consistent across all first-level feature groups. Notably, the clustering effect and the ring structure in the first-layer representations in the SORT tasks were more strongly observed in the attention-backload models (see Fig S8 in Appendix A.2). This could suggest that room for flexible multi-headed transformation downstream can relax first-layer encoding, enabling a more consistent solution to emerge across the GROUP and SORT tasks.

## 4 RELATED WORK

There is a growing interest in analyzing small models to better understand the capabilities and detailed computations in transformers in more controlled settings. For example, Power et al. (2022) explored learning and generalization dynamics in two-layer causal transformers, and Elhage et al. (2021) explored mechanistic interpretability in one- and two-layer transformers without MLP sublayers. Our work contributes to these efforts in beginning to shape some understanding of the computation and representation dynamics in small-scale transformers.

Ablation experiments and a variety of representation analyses have been applied to understand the role of the attention mechanism in NLP tasks as well as in transformer-based vision models (Chefer et al., 2021; Manning et al., 2020; Michel et al., 2019; Voita et al., 2019). However, it is often debated to what extent attention weights afford model interpretability in these settings (Jain & Wallace, 2019; Wiegreffe & Pinter, 2019; Vashishth et al., 2019), especially considering head redundancy and the difficulty in correctly attributing relevance over high-dimensional inputs. We show that at least in simple settings, the attention heads can exhibit some level of interpretability consistent with known task structures. Similar methods have also been applied to understand unit-level and layer-level dynamics that support multi-task computation in small-scale RNNs (Driscoll et al., 2022; Yang et al., 2019), which revealed some patterns that are consistent with our findings here, as we discuss below.

Recent work examining systematic generalization in transformers or pre-trained language models highlighted that length generalization remains a challenge and observed that positional encoding can have a significant impact on the models' ability to systematically generalize (Anil et al., 2022; Csordás et al., 2021a; Delétang et al., 2022; Ontanón et al., 2021). A concurrent work explored the randomized position encoding method (equivalent to our label-based encoding method) and showed that it outperforms a variety of sequence order encodings (Anonymous, 2022). Other types of architectural modifications have also been proposed to help transformers achieve better length generalization (e.g. Csordás et al., 2021b; Dehghani et al., 2018; Press et al., 2021; Su et al., 2021). Our work adds to the continued effort to enable better length generalization in transformers by demonstrating the potential in formulating sequence modeling tasks with a more general item-label binding approach rather than item-position binding.

Outside of the context of transformers and language models, the use of synthetic, algorithmic tasks has enabled much understanding of the core capabilities of many models (e.g. Graves et al., 2014). There has also been some interest in performing algorithmic reasoning with neural networks for its

own sake (Veličković & Blundell, 2021). Correspondingly, Veličković et al. (2022) recently proposed a benchmark for algorithmic reasoning, and evaluated a variety of graph neural network architectures, all of which struggled to extrapolate algorithms to longer or larger inputs.

## 5 DISCUSSION

We sought to understand how transformers can solve a set of highly-structured tasks and systematically generalize. We presented two-layer causal transformers that can learn copying, reversing, and hierarchical sorting operations and generalize to sequences longer than seen during training. We found that these models learned to exploit shared task structures and exhibited interesting signatures of task decomposition. The first layer tended to contextualize the input items with the task token or with other items in a feature group, and the subsequent layer was more responsible for within-group or task-specific processing.

Our findings on the contributions of individual attention heads in a multi-headed layer in solving these algorithmic tasks echo results from analyses of language models. For example, single attention heads are rarely responsible for a particular task or syntactic relationship and can often appear redundant (Manning et al., 2020; Michel et al., 2019; Voita et al., 2019). We do see some degree of selectivity, with conceptually distinct task components shared across a subset of attention heads. Recent work studying multi-task computation in RNNs has similarly found that multi-task learning led to the exploitation of reusable computation across related tasks (Driscoll et al., 2022; Yang et al., 2019). Interestingly, these recurrent models also exhibited high degrees of task-selectivity at the level of individual units in the hidden layer. The degree of task-selectivity in different architectural components may vary depending on the inductive biases of different model families and the tasks being learned. Future work is needed to fully characterize the degree of computational modularity in multi-headed attention in relation to task structures.

We highlight that the label-based order encoding method was key to enabling our models to generalize the learned tasks to longer sequences. The key insight is to sample random labels to communicate sequence order information instead of relying on sequence positions. This simple extension exposes models to a large range of possible labels evenly during training so they can learn to encode items in longer sequences with familiar labels. Compared to this approach, learnable position embedding leaves untrained embedding weights in unseen positions and thus would result in poor length generalization. However, learnable position encoding also resulted in slower learning on the training sequences in our tasks – potentially because the models had fewer sequences to learn to encode later positions compared to early ones. Sinusoidal position embeddings have been noted to have limited length generalization capabilities (Ontanón et al., 2021; Csordás et al., 2021a). In our setting, it may have additionally suffered from the demand to read out the absolute item positions, which is not common across NLP tasks. A concurrent work similarly noted the out-of-distribution problem with position-based encoding and demonstrated the superiority of the randomized position approach (equivalent to our label encoding method) for length generalization in a range of different algorithmic tasks (Anonymous, 2022). Although label-based encoding alone would fail to represent true item distance information, in light of recent work showing that transformers can learn positional information without explicit positional encodings (Haviv et al., 2022), it is promising that the advantages of label-based encoding may transfer to natural language inputs.

The flexibility to learn and perform multiple tasks is a key desired capability for machine learning. Our work provides insights into the possible structures of within-task and cross-task computations that stacks of attention layers can afford through a thorough analysis of the dynamics in a simple transformer on a set of highly-structured tasks. Recent work has explored multi-task learning in transformers at scale and achieved impressive results (Lee et al., 2022; Reed et al., 2022). As transformers are increasingly being leveraged for multi-task or multi-modal learning in domains with richer, more complex task structures, it is possible that these models may implicitly learn to decompose complex decisions into reusable, multi-level policies. In future work, we hope to explore these learning and generalization dynamics in transformer-based agents to understand the acquisition of task-conditioned, multi-level behavioral policies in structured environments.

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

# A  ADDITIONAL RESULTS

## A.1  SINGLE-TASK LEARNING

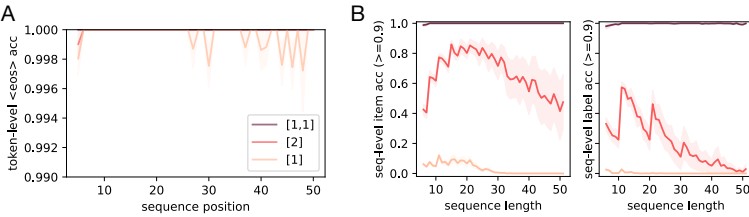

Figure S1: Additional teacher forcing accuracy. **A**. Single-task models predict EOS token near perfectly. **B**. Sequence-level accuracy when up to 10% errors were allowed. Results were taken from 5k training sequences (in B) and 5k generalization sequences (in A and B).

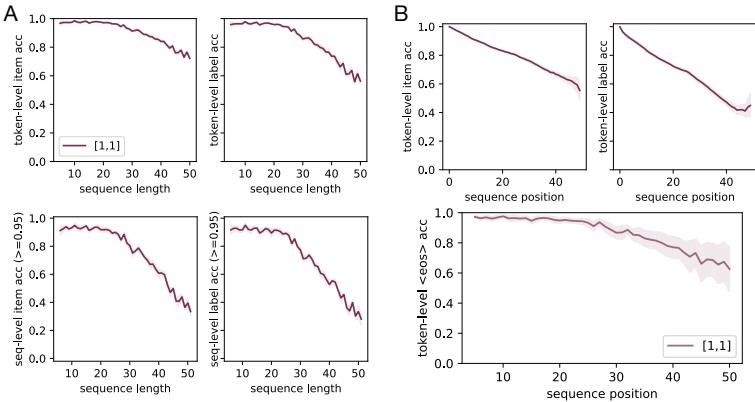

Figure S2: Top1 rollout accuracy for two-layer models. **A**. Token-level (upper) and sequence-level (lower) accuracy across sequence length. **B**. Token-level accuracy across sequence positions. Results were taken from 5k training sequences (in A) and 5k length generalization sequences (in A and B).

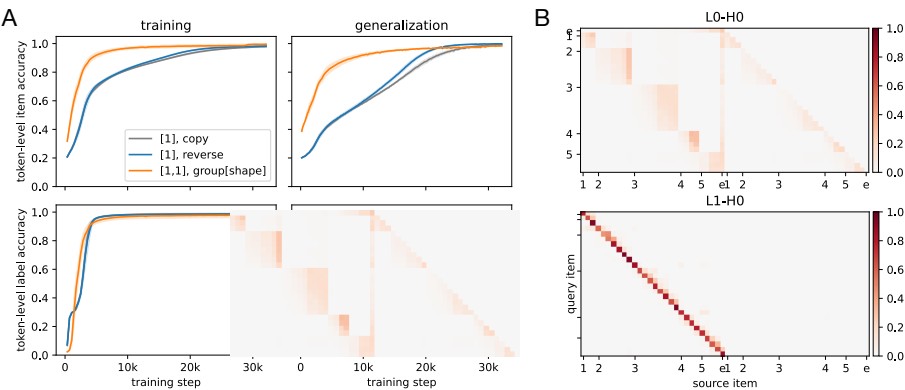

Figure S3: Learning other tasks in the single-task setting. **A**. Single-layer, single-headed models can learn the COPY task and the REVERSE task, while two-layer, single-headed models learn the GROUP[SHAPE] task. **B**. The attention maps from the GROUP[SHAPE] model resemble that from the SORT[SHAPE] model (visualization as in Fig 3A).

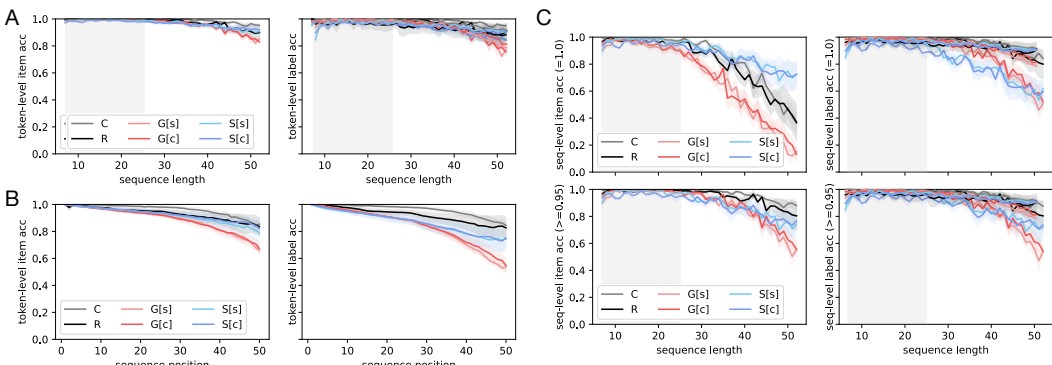

Figure S4: Attention maps from single-layer models in an example generalization sequence. **A**. Single-layer, single-headed model. **B**. Single-layer, two-headed model.

## A.2 MULTI-TASK LEARNING

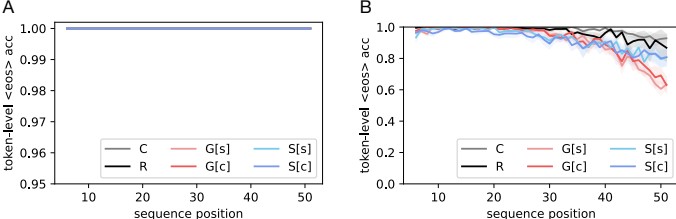

Figure S5: Visualization as in Fig 6, but item and label predictions were obtained using top1 rollout.

Figure S6: **A**. Six-task models predict the EOS token perfectly under teacher forcing. **B**. Prediction of the EOS token deteriorates under top1 rollout in six-task models. Results were taken from 1k training sequences and 1k length generalization sequences across the five top-performing runs.

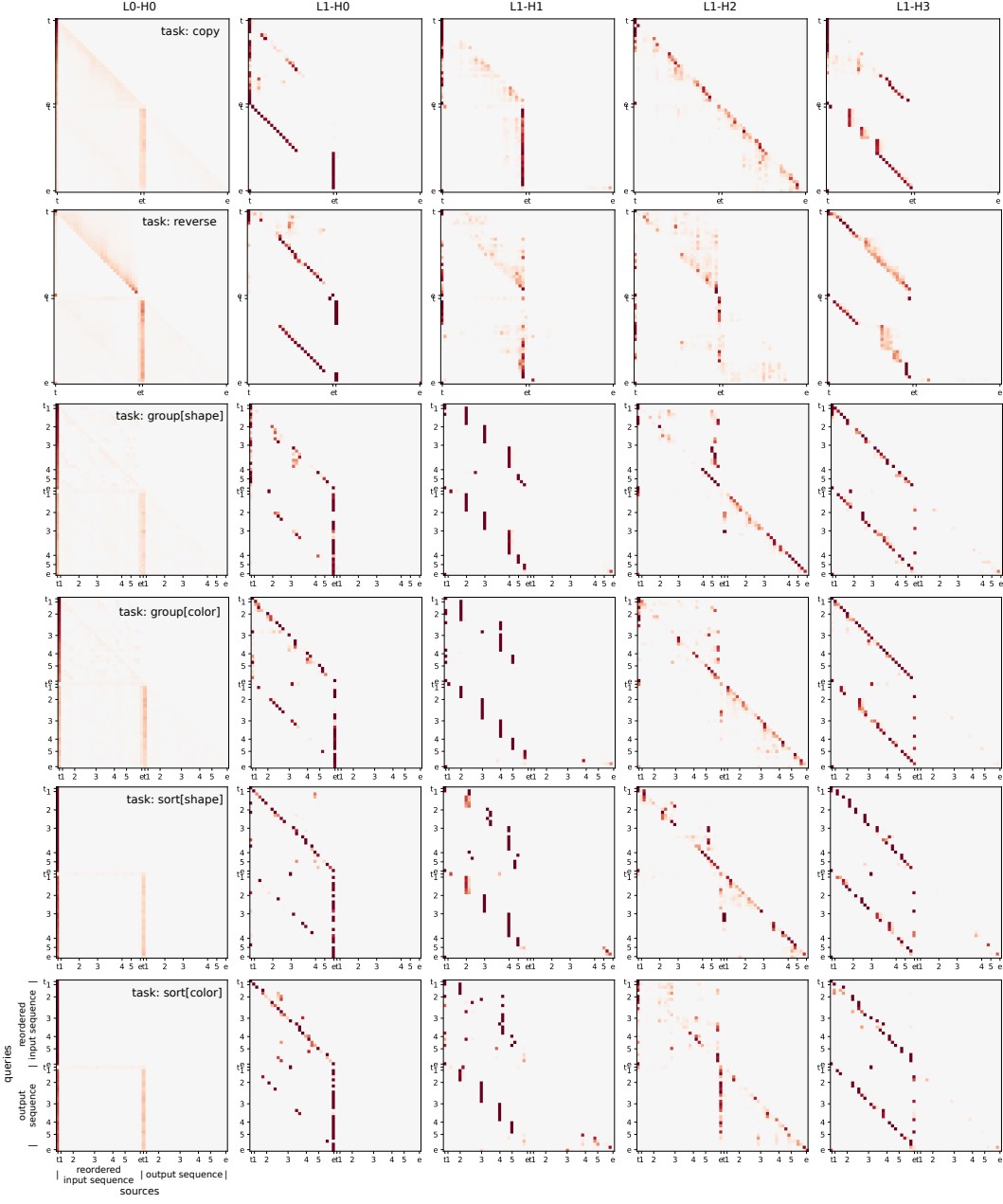

Figure S7: Attention maps for an example generalization sequence (for the top-performing multi-task model with [1,4] architecture). Each row shows the attention maps for one task, and each column corresponds to a single attention head denoted by the layer and head index. Tokens corresponding to the input sequence were reorderd to match their order in the output sequence for visualization purposes. Label t indicates the task token. Label e indicates the EOS token. In the bottom four rows, number labels ranging from 1-5 mark the beginning of the items in each first-level feature group.

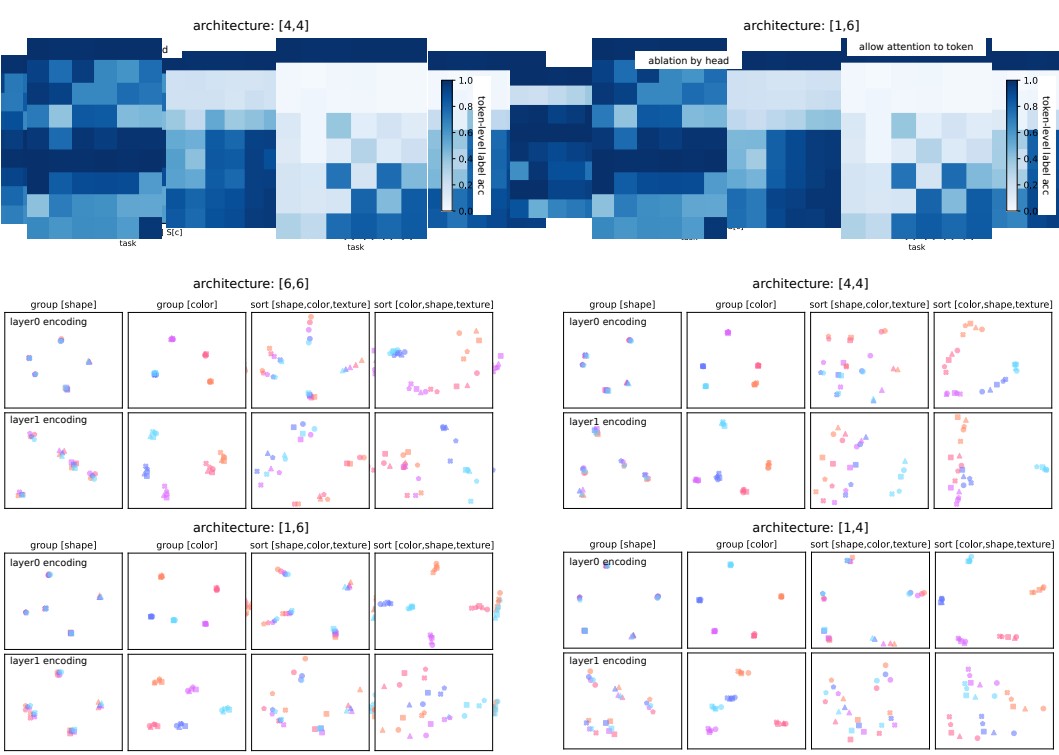

Figure S8: Attention ablation and task-conditioned item representations in other top-performing models. Visualization as in Fig 7B and Fig 8.

## B HYPERPARAMETERS

Table S1: Model and experiment hyperparameters.

| Hyperparameter | Single-task learning | Six-task learning |
|---|---|---|
| Number of layers | 1 or 2 | 2 |
| Number of attention heads | (see paper) | |
| Embedding dimension | 128 (two-layer) or 184 (single-layer) | 192 |
| MLP hidden layer size | 64 | |
| Activation function | ReLU | |
| Batch size | 128 | |
| Training teacher forcing rate | 1.0 | |
| Optimizer | Adam | |
| Learning rate | $10^{-4}$ | $5 \cdot 10^{-4}$ |

## C QUANTITATIVE PERFORMANCE

Table S2: Token-level item and label accuracy across 54k generalization sequences in single-task models learning the SORT [SHAPE,COLOR,TEXTURE] task. Mean and standard deviation across four random seeds are shown for each architecture.

| task | architecture | position encoding | item prediction | label prediction |
|---|---|---|---|---|
| C | [1] | label | 99.03±0.41 | 100.00±0.00 |
| R | [1] | label | 99.75±0.21 | 100.00±0.00 |
| S[s] | [1] | label | 76.49±2.99 | 51.75±5.35 |
| S[s] | [2] | label | 91.01±2.76 | 78.02±5.89 |
| S[s] | [1,1] | label | 99.60±0.07 | 98.41±0.23 |
| G[s] | [1,1] | label | 98.41±0.32 | 99.31±0.16 |
| S[s] | [1,1] | sinusoidal | 57.04±10.75 | 16.50±6.52 |
| S[s] | [1,1] | learnable | 73.15±11.65 | 40.51±14.28 |

Table S3: Token-level prediction accuracy across 75k generalization sequences in six-task models (including 500 unique sequences for each length evaluated within each task). Mean and standard deviation across four random seeds are shown for each architecture. Model architectures are denoted by [# heads in first layer, # heads in second layer].

(a) Attention-balanced models.

| Tasks | | [1,1] | [2,2] | [3,3] | [4,4] | [6,6] |
|---|---|---|---|---|---|---|
| all | item | 83.38±13.05 | 97.04±0.88 | 98.55±0.46 | 98.79±0.94 | **99.11±0.32** |
| | label | 87.21±7.86 | 96.46±1.24 | 97.98±0.42 | 98.68±0.84 | **99.17±0.32** |
| C | item | 89.28±14.44 | 97.57±1.19 | 98.81±0.77 | 98.98±0.95 | **99.13±0.97** |
| | label | 97.39±0.84 | 98.59±0.99 | 99.46±0.39 | 99.64±0.34 | **99.79±0.17** |
| R | item | 64.30±24.46 | 96.95±1.21 | 98.95±0.45 | 98.69±0.94 | **99.20±0.60** |
| | label | 94.56±8.26 | 98.53±0.80 | 99.64±0.16 | 99.39±0.29 | **99.73±0.19** |
| G[s] | item | 86.47±13.51 | 96.51±0.93 | 98.42±0.66 | 98.52±1.15 | **98.54±0.47** |
| | label | 91.54±8.30 | 97.83±1.07 | 99.07±0.33 | **99.23±0.52** | 99.05±0.30 |
| G[c] | item | 86.30±13.39 | 96.72±0.80 | 98.28±0.65 | 98.45±1.13 | **98.62±0.47** |
| | label | 91.69±7.79 | 97.93±1.06 | 98.98±0.19 | 99.21±0.46 | **99.22±0.24** |
| S[s] | item | 86.97±8.80 | 97.28±1.00 | 98.48±0.29 | 99.01±0.82 | **99.59±0.25** |
| | label | 74.75±11.47 | 93.21±2.23 | 95.53±1.01 | 97.09±1.99 | **98.62±0.82** |
| S[c] | item | 86.77±8.63 | 97.20±0.95 | 98.37±0.33 | 99.08±0.70 | **99.62±0.13** |
| | label | 73.57±10.59 | 92.73±2.05 | 95.23±0.90 | 97.53±1.71 | **98.65±0.54** |

(b) Attention-frontload models.

| Tasks | | [2,1] | [3,1] | [4,1] | [6,1] |
|---|---|---|---|---|---|
| all | item | 69.45±17.65 | 68.05±19.27 | 77.86±22.15 | **97.76±0.16** |
| | label | 79.95±8.53 | 78.35±11.60 | 86.53±11.67 | **97.52±0.31** |
| C | item | 57.93±27.09 | 54.07±30.36 | 72.01±30.48 | **98.50±0.81** |
| | label | 97.66±1.64 | 95.18±3.28 | 98.48±1.10 | **99.61±0.22** |
| R | item | 57.97±26.94 | 55.33±29.52 | 70.49±32.68 | **98.78±0.46** |
| | label | 95.74±3.34 | 94.90±3.50 | 97.97±2.17 | **99.56±0.23** |
| G[s] | item | 70.16±16.79 | 71.29±16.08 | 79.20±19.34 | **96.85±0.23** |
| | label | 80.56±11.79 | 79.37±12.51 | 87.82±12.52 | **98.96±0.26** |
| G[c] | item | 70.33±16.54 | 70.99±16.21 | 78.86±19.18 | **96.50±0.47** |
| | label | 79.18±12.71 | 78.71±12.82 | 87.28±12.69 | **98.82±0.32** |
| S[s] | item | 79.94±11.69 | 78.00±12.49 | 83.99±15.22 | **98.03±0.23** |
| | label | 62.76±13.85 | 61.17±19.50 | 76.69±18.13 | **94.17±0.74** |
| S[c] | item | 80.03±11.15 | 78.22±12.21 | 82.46±16.50 | **97.95±0.29** |
| | label | 64.28±12.87 | 61.27±18.82 | 71.27±24.19 | **94.08±0.82** |

(c) Attention-backload models.

| Tasks | | [1,2] | [1,3] | [1,4] | [1,6] |
|---|---|---|---|---|---|
| all | item | 96.06±0.24 | 98.65±0.80 | **99.40±0.29** | 99.29±0.17 |
| | label | 95.48±0.30 | 97.37±0.89 | **99.21±0.43** | 99.12±0.36 |
| C | item | 96.86±0.57 | 99.10±0.74 | **99.46±0.34** | 99.22±0.52 |
| | label | 97.78±0.71 | 99.62±0.24 | **99.85±0.08** | 99.59±0.55 |
| R | item | 95.99±0.59 | 98.82±1.08 | 99.47±0.37 | **99.56±0.14** |
| | label | 98.09±0.52 | 99.55±0.39 | **99.83±0.04** | 99.82±0.09 |
| G[s] | item | 96.12±0.38 | 98.28±1.00 | **98.98±0.43** | 98.96±0.16 |
| | label | 97.70±0.42 | 98.77±0.66 | **99.29±0.27** | 99.25±0.07 |
| G[c] | item | 95.82±0.72 | 98.53±0.76 | **99.17±0.34** | 98.78±0.40 |
| | label | 97.51±0.51 | 99.06±0.42 | **99.51±0.18** | 99.13±0.32 |
| S[s] | item | 95.86±0.77 | 98.51±0.72 | **99.68±0.26** | 99.59±0.13 |
| | label | 91.12±0.83 | 93.54±2.44 | 98.40±1.16 | **98.41±0.89** |
| S[c] | item | 95.72±0.54 | 98.70±0.71 | **99.64±0.28** | 99.63±0.29 |
| | label | 90.74±0.66 | 93.76±2.15 | 98.36±1.05 | **98.54±1.10** |

