# OpenReview forum: "Systematic Generalization and Emergent Structures in Transformers Trained on Structured Tasks"
_ICLR.cc/2023/Conference — Submitted to ICLR 2023_

### Official Review · Reviewer_ygnx · 2022-10-14

**Confidence:** 3
**Correctness:** 3
**Technical Novelty And Significance:** 3
**Empirical Novelty And Significance:** 3
**Recommendation:** 5

**Clarity, Quality, Novelty And Reproducibility:**

Clarity:

This paper is clear to read.
However, the words in the figures are too small.

Quality:

There are concerns about the task, data representation, and model designs (mentioned above).

Novelty:

There are novelties in proposing a new encoding method, analyzing with two-layer transformers and multiple findings.
However, whether they are still valid in more complicated cases is unclear.

Reproducibility:

The paper does not mention (anonymized) source codes.

**Strength And Weaknesses:**

***Strength***

- This paper proposes a simple yet effective new method that has improved over a state-of-the-art and widely used method.
- It analyzes how transformers address the generalization problem.
- It finds how to improve the performance (e.g., putting attention heads at deeper layers) and signs of systematic phenomenons.

***Weakness***

**The weakness mainly comes from the concern of whether the results and findings in the particular setting still hold in more complicated cases.**

**1. The advantage over conventional positional encoding**

The conventional positional encoding was designed for natural language inputs.
For example, it contains information on relative position or distance in input, as the context window is helpful for natural language.
The experiment in this paper does not have natural language input, and the tasks are not much related to relative position or distance in input.
So it is not very convincing that the advantage still exists in other tasks, e.g., natural language processing.

**2. Disentangled item representation**

The items contain shape, color, and texture.
However, the item units indicate the value in each feature dimension.
So the item representations are already disentangled, e.g., color and shape values are in different input nodes.
Addressing entangled data (e.g., the images of the items) is a critical expectation for systematic generalization.
The model behavior on disentangled data may not be the same as on entangled data.

**3. Model design**

It is not very convincing that findings in two-layer transformers will naturally extend to deeper transformers.
The limit of depth may prevent the model from having complicated behaviors.

**Summary Of The Paper:**

This paper proposes an encoding method that replaces the standard positional encoding in transformers. The label-based order encoding method achieves strong generalization to sequences longer than training ones.

This paper also analyzes two-layer causal transformers to learn multiple algorithmic operations. It finds that multi-level task learning improves when more attention heads at deeper layers.

The paper also finds that these models have signs of systematic decomposition within tasks and exploitation of shared structure across tasks.

**Summary Of The Review:**

This paper proposes a new encoding method that outperforms a widely used one and has informative analyses and findings.

However, the reviewer does not recommend acceptance because of the abovementioned weaknesses.

---

> ### Author Response · Authors · 2022-11-18
> **Response to Reviewer ygnx**
>
> Thank you for your feedback. To your point 1, as we mentioned in the overall response, we will point readers to a concurrent work comparing the same method against other methods. As for generalizing to natural language input, label-based encoding alone would not faithfully represent true token-distance information, which may affect the course of learning or performance in certain NLP tasks. It is worth noting that recent work showed that positional information seems to be learnable without any explicit position-related encodings (Haviv et al., 2022), thus it is possible that the advantages of label-based encoding can transfer to natural language inputs, though future work is needed to quantitatively confirm this.
>
> To your points 2 and 3, you’re right that the task set up already disentangled feature information and the model is very simple. Our main focus was in how generic self-attention can adapt to highly-structured tasks, and as a first step, it was encouraging to see that a simple causal transformer with no predefined task-aligned structure could learn clear structures in the data. In an image-based setting, due to the fact that any image can occur at any point in any sequence, the solution to the entanglement problem would naturally be placed in a shared encoding network that would be applied to all images, rather than inside the transformer layers. We note that we explicitly sought the minimal transformer that would reliably solve the simple task. In our view, it is of considerable value to understand the minimal solution, as this would likely provide a starting point for understanding more complex problems (e.g. those with the kinds of exceptions and partial regularities of natural data sets).  We’re definitely interested in exploring more complex task settings and more complex models in future work.

---

### Official Review · Reviewer_yZ2V · 2022-10-24

**Confidence:** 5
**Clarity, Quality, Novelty And Reproducibility:** The paper has limited novelty.
**Correctness:** 2
**Technical Novelty And Significance:** 1
**Empirical Novelty And Significance:** 1
**Recommendation:** 1

**Strength And Weaknesses:**

# Strengths
The paper provides a thorough analysis of a small Transformer on a set of algorithmic tasks.

# Weaknesses
1. The paper is of limited novelty. The major technical novelty is the proposed random label-based order encoding method. However, there lacks an explanation and insights why it works.

2. The authors do not conduct any experiment on the well established datasets, like SCAN. Besides, there is no baseline in this work. The authors should have comapred with Transformers using relative positional encoding (Csordas et al, 2021).

**Summary Of The Paper:**

In this paper, the authors explore Transformer's systematic generalization in algorithmic tasks, including copy, reverse, and hierarchical group or sort operations on an input sequence. The authors create a set of tasks and show that a two-layer Transformer successfully learns these tasks and generalizes to sequences longer. Particularly, a random label-based order encoding method, in place of the positional encoding, improves the systematic generalization of Transformer on the studied tasks.

**Summary Of The Review:**

The paper is of limited novelty and lacks comparison with previous methods. Therefore, I recommend rejection.

---

> ### Author Response · Authors · 2022-11-18
> **Response to Reviewer yZ2V**
>
> Thank you for your feedback. We have clarified why label-based order encoding would lead to better length generalization than position-based encoding methods in the revised paper. We were interested in how label-based encoding addresses the particular out-of-distribution problem position-based encoding methods encounter when only trained up to a max training sequence length L. Label-based encoding can sample encoding beyond L during training so that longer sequences at test time can be encoded with familiar labels. As such, we compared our method to two position-based encoding baselines. As we mentioned in the overall response above, we have pointed readers to a concurrent work comparing the same method against other methods in the revised paper.

---

### Official Review · Reviewer_1ohG · 2022-10-25

**Confidence:** 5
**Correctness:** 2
**Technical Novelty And Significance:** 3
**Empirical Novelty And Significance:** 3
**Recommendation:** 3

**Clarity, Quality, Novelty And Reproducibility:**

**Clarity**

The paper is generally well-written and easy to follow. However, several aspects of the paper are unclear:
* Is there a quantitative difference between multi-hot and one-hot encodings of the input sequence? In principle, the multi-hot encodings could be reformulated as one-hot encodings. To the best of my knowledge, using multi-hot encoding is quite nonstandard.
* How do the grouping and sorting tasks differ? Is the order of the grouped items irrelevant? If yes, how is that evaluated?
* Are the labels randomized at every training step or just once per sequence?
* Does the label-based encoding correspond to the original learned encoding for sequences of length 50, given that the item labels range only from 0 to 49?
* Why does the paper consider item *and* label prediction, which is quite nonstandard, as the paper states itself? Also, why does the model have to predict the task token in the multi-task setting if the task is provided as input?

Moreover, there are a few minor mistakes:
* Abstract: analysis -> analyses
* Figure 5: cross -> across

**Quality**

As mentioned in the weaknesses section above, the paper fails to discuss and compare to a series of highly related works, which calls into question the practicality and validity of the results.

**Novelty**

To the best of my knowledge, label-based encodings have not been proposed by prior work.

**Reproducibility**

The paper does not provide sufficient details of the experimental setup to reproduce the results. In particular, the sampling process used for label assignment is not specified. Code is not provided.


**Strength And Weaknesses:**

**Strengths**

The paper proposes a novel position embedding method that increases length generalization on a series of algorithmic tasks compared to learnable and sinusoid position embeddings. To that end, the paper introduces a new suite of algorithmic tasks and considers the single-task and multi-task settings, which are interesting. Moreover, the paper conducts an in-depth analysis of the model behavior on the different tasks and shows that, e.g., a two-layered transformer trained on the sorting task learns to first sort shapes and then the other features.

**Weaknesses**

The paper fails to discuss and compare to relevant related work:
* The paper does not compare its position embedding method to ALiBi encodings (Press et al., 2021), which were proposed to increase the length generalization capabilities of transformers on natural language processing tasks.
* The paper does not compare its position embedding method to the Neural Data Router (Csordás et al., 2022), which is an extension of the transformer architecture (consisting of shared layers, gating, geometric attention, and directional encodings) that significantly improves the systematic generalization capabilities of transformers on a wide range of algorithmic tasks.
* The paper does not discuss the length generalization benchmark introduced by Delétang et al. (2022), which evaluates transformers on a highly related set of tasks, including reversing, copying, and sorting. In particular, Delétang et al. (2022) assess the length generalization capabilities on significantly longer sequences (training on lengths up to 40 and evaluating on lengths 41 to 500). Given the accuracy decrease over sequence lengths (see Figures 2 D) and 6 C)), I am not convinced that the proposed encodings meaningfully increase length generalization for substantially longer sequence lengths.
* It would be interesting if the paper would discuss recent work showing that transformers can learn positional information without position embeddings (Haviv et al., 2022).

Moreover, while the paper shows that its proposed position embedding increases the transformer's length generalization on a set of algorithmic tasks, it does not evaluate its effectiveness on natural language processing tasks. In particular, it would be interesting to see if the proposed embedding can also be used to "train short and test long", as demonstrated by the ALiBi encoding (Press et al., 2021).


**Summary Of The Paper:**

The paper introduces a novel positional encoding that improves transformers' systematic (length) generalization capabilities on a set of algorithmic tasks, including copying, reversing, sorting, and grouping. Concretely, the paper replaces the learnable position embedding method with random labels from longer sequences, thus allowing the model to encode the longer sequences with familiar labels. The paper empirically demonstrates that the proposed method achieves better length generalization performance than learnable and sinusoidal position embeddings. Finally, the paper analyzes the attention matrices of the trained models and shows that they implement the algorithmic operations required to solve the corresponding tasks.

**Summary Of The Review:**

Given the insufficient comparison to related work, which calls into question the practicality and validity of the results, I do not recommend accepting the paper in its current form.

---

> ### Author Response · Authors · 2022-11-18
> **Response to Reviewer 1ohG**
>
> Thank you for your feedback. As we mentioned in the overall response above, we were interested in how label-based encoding addresses a particular limitation of position-based encoding. We have made this clearer in the revision and pointed the readers to a concurrent work that compares the same method to a range of other methods. As for generalizing to natural language input, label-based encoding alone would not faithfully represent true token-distance information, which may affect the course of learning or performance in certain NLP tasks. Given recent work showing that positional information seems to be learnable without explicitly appearing as a part of input embeddings (Haviv et al., 2022, as you noted), it is promising that the advantages of label-based encoding may transfer to natural language inputs, though future work is needed to quantitatively confirm this. We have refined these discussions in our revised paper.
>
> Please see our response below for your other questions:
>
> - Is there a quantitative difference between multi-hot and one-hot encodings of the input sequence? In principle, the multi-hot encodings could be reformulated as one-hot encodings. To the best of my knowledge, using multi-hot encoding is quite nonstandard.
>
> The multi-hot encoding of the items in the item pool is equivalent to one-hot encoding each individual item feature. We use the multihot setup to study how the model utilizes fully disentangled feature information in learning these tasks. If we tokenize each unique item with one-hot encoding, the same information is certainly learnable, but it could affect how quickly the model learns the latent features that are directly captured in the 3-hot encodings.
>
> - How do the grouping and sorting tasks differ? Is the order of the grouped items irrelevant? If yes, how is that evaluated?
>
> The group tasks make use of both input order and feature information to rearrange the items and are equivalent to a copy task within each feature group. The order of the feature groups is with respect to the same sort order as in the sort tasks (e.g., red ones go first, blue ones next, etc). The sort tasks further sort items in a feature group according to other features (e.g., sort by color within each shape, then sort by texture within each color).
>
> - Are the labels randomized at every training step or just once per sequence?
>
> They certainly can be sampled online for each incoming sequence during training or evaluation. In our case, we pre-generated a set of labels for each sequence in the dataset and these labels are shared across training steps and models.
>
> - Does the label-based encoding correspond to the original learned encoding for sequences of length 50, given that the item labels range only from 0 to 49?
>
> Yes, this is precisely the idea, in that we use the entire range of encodings up to length 50 for training sequences that only go up to length 25, so the model can learn to be familiar with encodings that would have been out-of-distribution for position-based encoding methods. Of course, exactly what range to sample the labels from is a hyper-parameter, which we set to the maximum generalization length in our setting.
>
> - Why does the paper consider item and label prediction, which is quite nonstandard, as the paper states itself? Also, why does the model have to predict the task token in the multi-task setting if the task is provided as input?
>
> You’re right that these are more particular to our task setup. We trained the models to predict both item feature and label information mainly because the tasks can use either (or both) feature and order information to rearrange the sequence, so we equate the actual target output requirements across tasks. Predicting the task token is less important. We chose to implement it this way for potential test cases of multi-task operations on one sequence.

---

### Author Response · Authors · 2022-11-18
**General Response**

We thank all reviewers for taking the time to review our submission and providing valuable feedback. One common theme that they raise is the lack of comparison of our proposed label-encoding method to previous methods such as ALiBi and relative positional encoding, as well as a demonstration of the effectiveness of this method on other datasets/tasks. These are important questions, and we have considered taking things in this direction. We want to clarify that the principled focus of this work was to explore how self-attention can adapt to clear structures in the data by analyzing the representations and attention patterns when transformers learn algorithmic tasks. We came across the idea of label-based encoding in the course of doing the work, and were interested in how it addresses the particular out-of-distribution problem associated with position-based encodings on sequences longer than seen during training. We have made this motivation clearer in the revised paper. We also note that after submission, we became aware of a concurrent work that studied this problem and compared the label-encoding approach to numerous other methods. We’re glad to see that this idea has also been engaged by other people, and will refer the readers to that work for comparisons to other methods.

---

### Decision · Program_Chairs · 2023-01-20

**Decision:**

Reject

**Justification For Why Not Higher Score:**

Several of the reviewers pointed out that both the models and the tasks considered here are relatively simple, which makes it difficult to assess whether the insights here would also hold for more realistic applications of Transformers.

**Justification For Why Not Lower Score:**

N/A

**Metareview: Summary, Strengths And Weaknesses:**

This paper studies the length generalization abilities for Transformers on algorithmic tasks, particularly when a new learnable position embedding method with random labels is used in place of relative position encodings. Both length generalization and Transformer's performance on algorithmic tasks are interesting and timely topics to study. However, several of the reviewers pointed out that both the models and the tasks considered here are relatively simple, which makes it difficult to assess whether the insights here would also hold for more realistic applications of Transformers.